# Fronthaul Design for Wireless Networks

**Ivo Sousa *** , **Nuno Sousa, Maria Paula Queluz** and **António Rodrigues**

Instituto de Telecomunicações, IST, University of Lisbon, 1049-001 Lisbon, Portugal;
nunoguerreirodesousa@gmail.com (N.S.); paula.queluz@lx.it.pt (M.P.Q.); antonio.rodrigues@lx.it.pt (A.R.)
* Correspondence: ivo.sousa@lx.it.pt; Tel.: +351-218-418-454

**Abstract:** Cloud Radio Access Network (C-RAN) architectures have arisen as an alternative to traditional wireless network architectures, notably by taking advantage of the functional split between the multiple distributed Remote Radio Heads (RRHs) and the centralized Baseband Units (BBUs), through the creation of a new connectivity segment—the fronthaul. In order to maximize the investment return, it is important to find out, for this C-RAN segment, which technologies provide cost-effective solutions. This paper addresses this issue by evaluating and comparing the performance of Microwave Radio Transmission (MRT), Free Space Optics (FSO), and Fiber Optics (FO) technologies when applied to the fronthaul. First, a methodology is provided to determine the most cost-effective solution for each RRH–BBU link, as well as to compute the required number of BBUs and where they should be positioned in order to minimize the overall network costs. Next, a cost-effectiveness comparison of the aforementioned communication technologies is presented for individual fronthaul segments under different weather conditions, link lengths, and bit rate requirements. Moreover, an assessment is performed regarding the impact of the RRH density on the selection of cost-effective communication technologies for C-RANs. The obtained results allow concluding that fronthaul expenses are significantly affected by the performance of FSO systems, which in turn is affected by weather conditions; this highlights the relevance of having accurate climate statistics and forecasts in order to get the most out of the FSO technology and, consequently, lowering the overall network costs.

**Keywords:** Cloud Radio Access Network (C-RAN); fronthaul; Microwave Radio Transmission (MRT); Free Space Optics (FSO); Fiber Optics (FO)

---

## 1. Introduction

Mobile phones are currently the most-often used means of accessing the Internet, and the number of mobile communication subscriptions continues to grow [1]. This results from the fact that consumers now expect to access data when and where they need them, as well as to have higher throughput while paying the same (or even less) in the future. On the other hand, the high consumption of mobile data represents a critical revenue driver for operators [2]; in fact, as data traffic becomes more and more dominant over time, the revenues become flat [3], thus putting an immense amount of pressure on the mobile network providers, which need to optimize costs while providing high-capacity and high-quality services to their subscribers.

It is generally recognized that an increase of the mobile cell density is an effective way to attain vastly high data rate coverage. However, the respective deployment costs may hinder the Radio Access Network (RAN) improvement; hence, cost-effective solutions are needed for this part of the mobile telecommunication system, so that the investments can be supported by the revenues. Mobile network virtualization, notably the concept of Cloud-RAN (C-RAN) [4,5], arose as a path to lower Capital Expenditures (CAPEX) and Operational Expenditures (OPEX), as it enables the sharing of infrastructures, which can help to reduce the overall expenses of deployment and operations.

Nevertheless, the costs of fronthauling the mobile cells—i.e., the expenses to establish connections between the multiple distributed Remote Radio Heads (RRHs) and the centralized Baseband Units (BBUs)—still pose an important challenge.

This paper considers the C-RAN concept and aims at finding out, for a wide range of scenarios, which underlying fronthaul technologies are cost-effective; the key technologies examined are Microwave Radio Transmission (MRT), Free Space Optics (FSO), and Fiber Optics (FO). The following contributions are made by this paper: (1) a cost-effectiveness comparison of the aforementioned communication technologies is performed for individual fronthaul segments under different weather conditions, lengths of the links, and bit rate requirements; (2) an assessment is also performed regarding the impact of the RRH density on the selection of cost-effective communication technologies for C-RANs. These studies were made possible by the development of a methodology that enables determining the most cost-effective solution for each RRH–BBU link, as well as to compute the required number of BBUs and where they should be positioned in order to minimize the overall network costs. In a nutshell, the provided methodology along with the findings of this work aim at serving as useful guidelines when designing fronthaul networks.

This paper is organized as follows. After the introduction, Section 2 provides a background on the topics of interest for this work, namely C-RAN and fronthaul communication technologies, as well as a review of related work. Section 3 describes the developed methodology to evaluate and compare the performance of MRT, FSO, and FO technologies when applied to the fronthaul. Section 4 provides and discusses the results of the studies performed herein, namely an evaluation of which technology (MRT, FSO, or FO) is the most cost-effective choice for an individual RRH–BBU link regarding the conditions of its implementation, as well as a cost-effectiveness analysis of communication technologies for C-RANs with different RRH densities. Finally, Section 5 concludes the paper.

## 2. Background on C-RAN and Fronthaul Communication Technologies and Related Work

In order to contextualize the topics address in this paper, this section presents a description of the C-RAN architecture, which enables pointing out the role of the fronthaul segment within the mobile network. Moreover, typical fronthaul communication technologies are also overviewed and compared, namely MRT, FSO, and FO. The section ends with a review of related work.

### 2.1. Cloud Radio Access Network and Fronthaul

In its traditional architecture, the RAN comprises costly stand-alone Base Stations (BSs), each one serving a certain area and thereby only handling transmitted and received signals of the User Equipment (UE) within that same area. In order to attain high data rate coverage, a large number of BSs is required, which also poses some challenges, such as the costly initial investment, site rental, and site support; thus, an increase in the number of BSs gives rise to a significant increase in CAPEX and OPEX. On the other hand, traditional BSs are configured to fulfill peak demands in order to reduce outages; however, since the average network load is usually far lower than that in peak load, the BS utilization rate ends up being low. Noticing that the processing capacity of each BS cannot be shared with others under the traditional RAN architecture, reducing the number of BSs or their processing resources is not an option to optimize their utilization efficiency, as it would lead to many and unacceptable congestion events during peak hours, or even a lack of coverage issues.

In order to reduce the overall expenses of deployment and operations, the C-RAN architecture was proposed: a radio access network model in which baseband resources are pooled so that BSs can share them. This is achieved by separating the traditional BS into an RRH, which is responsible for power amplification and analog processing, and a BBU, which manages the common digital baseband functions, such as modulation and coding. The optimization of the BBU usage is reached through the virtualization of a BBU pool, which can then be shared among cell sites, i.e., RRHs. Figure 1 depicts a typical C-RAN architecture, which highlights the backhaul segment, i.e., the connection between the core network and the BBU, and also the new connectivity segment—the fronthaul—introduced

by the C-RAN architecture between the multiple distributed RRHs and the BBU. According to the related literature [4,5], a C-RAN presents the following benefits relative to a traditional RAN: energy efficiency, decrease in CAPEX and OPEX, throughput improvement, adaptability to non-uniform traffic, and smart Internet traffic offload.

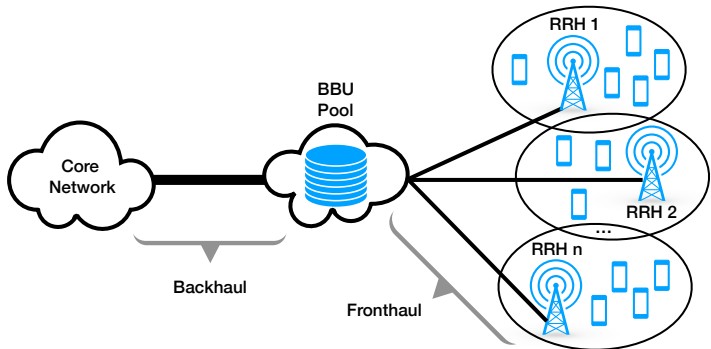

**Figure 1.** C-RAN architecture. BBU, Baseband Unit; RRH, Remote Radio Head.

One of the main technical challenges regarding the C-RAN architecture is the large amounts of baseband sampling data that have to be carried out in real time between RRHs and the BBU pool. Besides strict latency requirements, the wideband required by some radio access systems, like Long-Term Evolution (LTE) or 5G, implies that each fronthaul link should support substantial bit rates, e.g., 1 Gbps [6].

### 2.2. Typical Fronthaul Communication Technologies

The adopted technologies for the fronthaul links and associated transmission medium, which can be wired or wireless, must be able to fulfill the requirements highlighted previously. Typically, the fronthaul segment uses terrestrial communication technologies, namely MRT, FSO, and FO—for instance, the MRT technology was considered as an option for the fronthaul in [4–7], whereas FSO systems were considered in [8,9], and the FO technology was contemplated in [4–7,10,11].

MRT is a line-of-sight wireless communication technology that uses microwave radio waves in order to provide high speed wireless connections, which generally can go up to several Gbps [12]. MRT is widely used for point-to-point communications, as the small wavelength allows the use of conveniently-sized antennas to create highly directional beams, making their installation rather simple and suitable for any terrain. Moreover, since the associated equipment can be easily disassembled, relocated, and reused, MRT systems are also a good easy-to-deploy option in the case of a catastrophe or emergency, especially for remote locations. Multi-path fading is a significant factor degrading the performance of MRT links operating below 10 GHz; however, when considering frequencies above 10 GHz, weather conditions represent the dominant cause of communication disruption, notably rain (which becomes a crucial attenuation factor for this technology) and atmospheric absorption [13]. It is also worth mentioning that relay stations may be used to overcome the line-of-sight limitation of MRT systems (notably for link lengths longer than the visual horizon). The installation costs of MRT systems are independent of each hop length (i.e., the distance between two MRT stations) if the same equipment is used; on the other hand, the total costs of an MRT link comprising multiple hops is greatly dependent on the number of hops used [14]. Furthermore, and in general, MRT systems operate in licensed frequency bands; hence, individual MRT systems are subject to license (on a case-by-case basis to avoid interference between adjacent systems) by the respective national regulatory authority for communications, which normally entails the payment of an annual fee (an operational cost that is usually square-root dependent on the link length) [15,16].

The FSO technology uses infrared radiation in free space (i.e., air, outer space, or vacuum) to wirelessly transmit data; for that reason, there has been growing recognition in recent years of the importance of FSO systems in the ongoing development of fixed telecommunication systems,

mainly because not only FSO links can be used in conjunction with MRT links without causing interference, but also the FSO technology is well suited for environments where the radio spectrum is already crowded (e.g., dense urban areas). In particular, FSO is a point-to-point line-of-sight communication technology like MRT, but it allows higher capacities, namely connections of up to several dozen Gbps [17]. An FSO system consists of optical transceivers at the stations; it has a low initial investment, and it is easy to deploy (similar to MRT systems). One characteristic of the FSO technology is the low diffraction of the laser generated beam; on the one hand, this provides high security of the transmitted data, as an interceptor would have to be perfectly aligned with the transmitter in order to obtain the data; on the other hand, the very narrow optical beam makes FSO a sensitive system, as physical obstructions (like trees or even flying birds) can block the signal and cause interruptions in a link. A major limitation associated with FSO links is the attenuation caused by fog, which can reach values as high as 480 dB/km in extreme situations [18]. Other attenuation factors are atmospheric turbulence, i.e., changes in the atmosphere refractive index (e.g., due to high temperatures), which can cause deviations in the beam, and interference from background light sources (including the Sun). The deployment costs of FSO systems are independent of each hop length (if the same equipment is used). Moreover, these systems have the advantage of being license free, and no planning permission is required as long as they are eye safe. Nevertheless, the maximum achievable range per hop is lower than in MRT [17].

FO communication systems are light wave systems that employ optical fibers for information transmission, which means that these are not wireless systems (as opposed to MRT and FSO technologies) and, therefore, do not require line-of-sight. The capacity of current optical fibers can reach values up to several hundred Gbps (per wavelength channel) [19], being mainly conditioned by optical pulse widening due to fiber dispersion, which limits the maximum bit rate × distance product (a metric that describes the fiber performance in terms of transmission capability). Since the optical fiber cable is usually installed underground, FO systems may be damaged by floods. Moreover, accessibility issues regarding the path of installation of optical fibers, such as those in mountainous regions, might lead to FO being regarded as an unsuitable option. The deployment costs of an FO system vary linearly (roughly) with the considered connection length [14]. Accordingly, the installation of FO systems is usually more expensive than MRT and FSO systems' installation.

A summary of the main differences between MRT, FSO, and FO is presented in Table 1. As can be inferred from this table, the best choice regarding the fronthaul communication technology is not straightforward, as it depends on the required bit rate, terrain characteristics, and typical weather conditions of the installation sites, among others.

**Table 1.** Comparison of typical fronthaul communication technologies.

|  | MRT | FSO | FO |
|---|---|---|---|
| **Type** | Wireless (requires line-of-sight) | | Wired |
| **Capacity** | Up to several Gbps | Up to several dozen Gbps | Up to several hundred Gbps |
| **Installation** | Easy to deploy and suitable for any terrain | | Unsuitable for difficult terrain (e.g., mountains) |
| **Climate influence** | Notably rain | Notably fog | Not influenced by climate (except floods) |
| **Other** | MRT maximum range per hop is higher than FSO | | Installation is usually more expensive |

## 2.3. Related Work

Since the fronthaul optimization can lead to a large decrease of CAPEX and OPEX of the C-RAN, it is of utmost importance to study and evaluate different approaches regarding the fronthaul architecture. The main objective of this paper is to evaluate and compare the use of MRT, FSO, and FO systems in the fronthaul, namely to study their influence in terms of cost-effective solutions for RRH–BBU links.

There are few works, available in the related literature, that tackle topics similar to the one of this paper. The authors of [20] analyzed operational and deployment costs of C-RANs and proposed a mathematical formulation for fronthaul planning, taking into account the aforementioned costs. Solutions were also presented regarding their optimization problem, which aim at minimizing the C-RAN costs subject to traffic demand constraints. Unfortunately, this work was limited in the sense that it considers only a single technology (namely FO) for the fronthaul.

In [21], different fronthaul technologies were discussed, including MRT, FSO, and FO, and potential solutions were suggested in order to achieve an efficient C-RAN. Nonetheless, the authors only provided a qualitative costs comparison of the different fronthaul technologies, without considering any specific scenario, as they focused their study on optical technologies (FSO and FO) in order to present feasible means of reducing the system complexity, costs, bandwidth requirement, and latency in the fronthaul.

A comparative costs study regarding the technologies addressed herein (MRT, FSO, and FO) along with networked flying platforms providing FSO links ("vertical FSO") was performed in [22] for a fronthaul/backhaul scenario. However, only a very high dense deployment of mobile cells was considered, and the technologies were compared in terms of the whole network costs when a single technology was adopted. Nevertheless, it can be concluded from their work that although the "vertical FSO" was a valid approach, it was a very expensive option—in fact, not only "vertical FSO" was shown to be the most expensive option, but also the associated costs were estimated to be more than twice the second most expensive solution; furthermore, its implementation posed additional problems, including safety and regulatory ones. For these reasons, the "vertical FSO" approach was not considered in our study.

Recently, other authors addressed rural connectivity and investigated the costs of solutions for the backhaul segment [23], namely by considering MRT, FSO (including the "vertical FSO" approach), and FO, in addition to satellite. In this case, the studied scenario is very specific—remote rural areas where the backhaul link needs to traverse long distances (e.g., 100 km); hence, its results cannot be generalized to the more common fronthaul scenario of RRH–BBU links separated by a few kilometers. Nevertheless, it could be concluded from their work that the "vertical FSO" solution was, again, more costly than the other options; on the other hand, their work showed that the cost-effectiveness of the MRT and FSO solutions depended, among other things, on the tower separation (namely, they compared hops of 3 km vs. 5 km ones), thus suggesting that a thorough study of this costs dependence on the hop distance, notably considering different equipment and communication technologies, would be of great value.

In light of the above, this paper advances the state-of-the-art by providing a quantitative cost-effectiveness comparison of three communication technologies, namely MRT, FSO, and FO, for individual fronthaul segments under different weather conditions, link lengths, and bit rate requirements. Furthermore, an assessment is also provided regarding the impact of low and high RRH densities on the selection of cost-effective communication technologies for C-RANs. In addition to the studies presented and discussed herein, this paper provides a generic and versatile methodology to determine the most cost-effective solution for each RRH–BBU link, as well as to compute the required number of BBUs and where they should be positioned in order to minimize the overall network costs.

## 3. Fronthaul Design Methodology

This section details the developed methodology to evaluate and compare the performance of MRT, FSO, and FO technologies when applied to the fronthaul. This methodology comprised a software tool built from scratch (based on the MATLAB programming language), which had the following purposes: (i) to determine the most cost-effective solution to connect two points regarding an RRH–BBU link, under user-specified equipment characteristics and link conditions; (ii) to find a fronthaul topology for wireless networks optimally, given the RRHs location—namely, the required number of BBUs and where they should be positioned in order to minimize the overall network costs. The algorithms associated with these goals, designated as the link design algorithm and network planning algorithm, respectively,

are presented next. The software tool, including its source code, is freely available online [24]; hence, besides being a generic tool, since it can be used for a wide range of fronthaul projects, it is also a versatile tool, in the sense that it can be modified by anyone to incorporate other features that are specific for a certain project.

### 3.1. Link Design Algorithm

The link design algorithm determines, out of all the user-specified equipment characteristics associated with one or more of the communication technologies (MRT, FSO, and FO), which is the most cost-effective solution to connect two points under user-specified link conditions; for a set of specifications, namely the equipment features and other installation aspects (such as link distance, required bit rate, and surrounding environment characteristics), this algorithm computes the cheapest solution that is able to deliver the necessary bit rate, while satisfying a certain link margin and error criteria.

With respect to the wireless technologies addressed herein (MRT and FSO), it is important to mention that only single-hop links are considered by the algorithm, because the inclusion of relay stations is not straightforward (e.g., sites for relay deployment may not be available) and leads to more complex business models (e.g., the addition of rental expenses for the extra sites). Accordingly, link distance is henceforth regarded as hop distance when MRT and FSO technologies are under consideration.

The following subsections present the details of the link design algorithm, namely the adopted models for the communication technologies, the economic analysis methodology, and finally, the workflow of the algorithm.

### 3.1.1. Communication Technologies Models

For wireless communication technologies such as MRT and FSO, the received signal power, $P_{Rx}$, is given by (in logarithmic units):

$$P_{Rx} = P_{Tx} + G_{Tx} + G_{Rx} - A_0 - A_{equi} - A_{sys}, \tag{1}$$

where $P_{Tx}$ corresponds to the transmitted power (in dBW or dBm), $G_{Tx}$ and $G_{Rx}$ stand for the transmitter and receiver antenna gains (in dBi), respectively, and $A_0$ represents the free-space path loss (in dB), i.e.,

$$A_0 = 92.4 + 20 \log_{10}(d_{[km]}) + 20 \log_{10}(f_{[GHz]}), \tag{2}$$

where $d$ denotes the link distance and $f$ refers to the carrier frequency, $A_{equi}$ denotes the losses (in dB) related to equipment like cables, modulators, etc. (which are typically lower than 3 dB), and $A_{sys}$ corresponds to other losses (in dB) related to specific attenuation factors regarding the considered communication technology.

With respect to MRT systems, the term $A_{sys}$ incorporates the attenuation caused by obstacles in the line-of-sight path ($A_{obs}$), the attenuation induced by atmospheric gases ($A_{gas}$), and the attenuation due to rain ($A_{rain}$). All these attenuation factors can be computed as described in the related literature [13]; a summary is given as follows. The attenuation due to obstacles can be computed as (in dB):

$$A_{obs} = \max \left\{ 0 \; ; \; 6.9 + 20 \log_{10}(\sqrt{(\nu - 0.1)^2 + 1} + \nu - 0.1) \right\}, \tag{3}$$

where $\nu$ is proportional to the amount of the Fresnel ellipsoid that is obstructed by the obstacle, i.e.,

$$\nu = \frac{h_{obs}}{17.32} \sqrt{8 \frac{f_{[GHz]}}{d_{[km]}}}, \tag{4}$$

where $h_{obs}$ denotes the height of the top of the obstacle above the straight line joining the two ends of the link (if the height is below this line, then $h_{obs}$ is negative). The attenuation due to atmospheric gases (namely uncondensed water vapor and oxygen) can be computed as (in dB):

$$A_{gas} = (\gamma_o + \gamma_w) \times d,$$ (5)

where $\gamma_o$ and $\gamma_w$ represent, respectively, the attenuation caused by oxygen and water by units of length; values for these parameters can be extracted from nonlinear curves as a function of the carrier frequency. With respect to $A_{rain}$, it should be pointed out that since rain is highly variable over time and differs from place to place, the respective attenuation factor depends on the desired time availability for the MRT link. More specifically, the attenuation due to rain has to be computed taking into account the value of rain intensity that is not exceeded in a certain percentage of the time in the location of interest. Accordingly, and having in mind the Service Level Requirements (SLR)—which address service times, maintenance, availability, performance, etc.—an MRT system designer must first stipulate the minimum time availability of the link (e.g., 99.9% of the time, as adopted in this work); afterwards, by using the method suggested by an ITU-R (International Telecommunication Union – Radiocommunication Sector) recommendation [25], the rain attenuation is computed, thus ensuring that the planned MRT link may still suffer from rain outage, but no longer than the maximum percentage of time unavailability of the link ($U_{max}$), with $U_{max} = 100\%$ − minimum link availability percent. Formally and given the rain intensity not exceeded in 0.01% of the time ($R_i^{(0.01\%)}$), the rain attenuation for that percentage of the time is given by (in dB):

$$A_{rain}^{0.01\%} = \beta \times \left( R_i^{(0.01\%)} \right)^{\alpha} \times d_{ef},$$ (6)

where $\beta$ and $\alpha$ denote coefficients that depend on the carrier frequency and on the considered temperature and $d_{ef}$ corresponds to the effective distance through a rainy path, which is computed as:

$$d_{ef} = \max \left\{ 2.5\, d\, ; \; \frac{d}{0.477\, d_{[km]}^{0.633} \times \left( R_i^{(0.01\%)} \right)^{0.073\,\alpha} \times f_{[GHz]}^{0.123} - 10.579 \left( 1 - e^{-0.024\, d_{[km]}} \right)} \right\} ;$$ (7)

finally, the rain attenuation exceeded for a percentage of time $U_{max}$ other than 0.01% is obtained as:

$$A_{rain} = A_{rain}^{0.01\%} \times 0.12\, U_{max}^{-(0.546 + 0.043 \log_{10} U_{max})}.$$ (8)

Turning the attention now to FSO systems, the term $A_{sys}$ encompasses the attenuation induced by atmospheric absorption ($A_{abs}$), the attenuation due to atmospheric turbulence ($A_{turb}$), and the attenuation caused by scattering ($A_{sca}$). All these attenuation factors can be computed as described in the related literature [26,27]; a summary is given as follows. Considering the atmospheric absorption (which is mainly caused by the presence of gaseous molecules), it can be given by (in dB):

$$A_{abs} = \gamma_{abs} \times d,$$ (9)

where $\gamma_{abs}$ stands for the absorption attenuation coefficient, which depends on the considered temperature and on the relative humidity. The attenuation due to atmospheric turbulence (i.e., small and random variations of the refractive index of the Earth's atmosphere, which are responsible for wave front distortion) can be described by (in dB):

$$A_{turb} = 2\, \sigma_{scin},$$ (10)

where $\sigma_{scin}$ refers to the scintillation index, which can be expressed as:

$$\sigma_{scin} = \sqrt{1.23 \, C_n^2 \times \left(\frac{2\pi}{\lambda_{[m]}}\right)^{\frac{7}{6}} \times d_{[m]}^{\frac{11}{6}}}, \tag{11}$$

where $\lambda$ corresponds to the operating wavelength and $C_n^2$ represents the index of refraction structure parameter, which is computed as (in m$^{-2/3}$):

$$C_n^2 = 9.8583 \times 10^{-18} + 4.9877 \times 10^{-16} \times e^{-\frac{h_{a[m]}}{300}} + 2.9228 \times 10^{-16} \times e^{-\frac{h_{a[m]}}{1200}}, \tag{12}$$

where $h_a$ denotes the transmitter altitude. With respect to $A_{sca}$, which is caused by the occasional presence of fog (including mist and haze) and rain, it is important to stress that fog is the major contributor regarding the attenuation due to scattering; hence, and noticing that the respective attenuation coefficient is computed as a function of the visibility, this type of attenuation also depends on the desired time availability for the FSO link. More specifically, one has to take into account the value of the visibility that is not exceeded for a given percentage of the time in the location of interest. Since the ITU-R recommendations for FSO systems' design do not provide a metric to compute the visibility distribution, one alternative is to use one of the visibility distribution models presented in a related work [28]—e.g., the simplified model introduced therein and adopted in this methodology—which rely on the average number of foggy days (per year) and the average duration of fog events (in hours). Accordingly, the visibility value can be obtained and used to compute the respective attenuation that ensures that the planned FSO link may still suffer from fog outage, but no longer than the maximum percentage of the time regarding tolerated link unavailability, thus enabling fulfilling the SLR; it is important to mention that, in order to ensure the fulfillment of the same SLR regardless of the specific adopted wireless technology, the software tool considers the same percentage of the time for link unavailability regarding the attenuations related to both rain and fog. Formally, the attenuation caused by scattering is given by (in dB):

$$A_{sca} = (\gamma_{fog} + \gamma_{rain}) \times d, \tag{13}$$

where $\gamma_{fog}$ and $\gamma_{rain}$ represent, respectively, the attenuation caused by fog and water by units of length. The former parameter can be expressed as (in dB/km):

$$\gamma_{fog} = \frac{3.91}{V_{[km]}} \times \left(\frac{\lambda_{[nm]}}{550}\right)^{-q}, \tag{14}$$

where $V$ stands for the visibility and $q$ refers to a coefficient that is dependent on the size distribution of the scattering particles, which is given by:

$$q = \begin{cases} 1.6 & V > 50 \text{ km} \\ 1.3 & 6 \text{ km} < V < 50 \text{ km} \\ 0.16 \, V_{[km]} + 0.34 & 1 \text{ km} < V < 6 \text{ km} \\ V_{[km]} - 0.5 & 0.5 \text{ km} < V < 1 \text{ km} \\ 0 & V < 0.5 \text{ km} \end{cases}. \tag{15}$$

With respect to the visibility, it can be computed as (in km):

$$V = \frac{U_{max}}{100} \times \frac{365.25}{N_{fog[days/year]}} \times \frac{24}{\overline{D}_{[h]}}, \tag{16}$$

where $\overline{N_{fog}}$ and $\overline{D}$ refer to the average number of foggy days and the average duration of fog events, respectively. Finally, the parameter related to the attenuation caused by water can be expressed as (in dB/km):

$$\gamma_{rain} = 1.076 \left( R_i^{(0.01\%)} \right)^{0.67} \times 0.12 \, U_{max}^{-(0.546 + 0.043 \log_{10} U_{max})}. \tag{17}$$

After computing the received signal power, the wireless link margin, $W_{link}$, is obtained as (in dB):

$$W_{link} = P_{Rx} - S_{Rx}, \tag{18}$$

where $S_{Rx}$ refers to the sensitivity of the receiver. In order to consider a wireless connection as viable and since the higher the link margin, the more robust the wireless link will be, as it will be prepared for potential extra attenuations, $W_{link}$ must be greater than a user-specified minimum accepted link margin (e.g., 3 dB, as adopted in this work for both MRT and FSO links).

Once the link margin requirement is satisfied, another requirement, namely the Bit Error Rate (BER), must be fulfilled in order to ensure that the wireless link is feasible (e.g., in this work, the BER must be lower than $10^{-6}$). The BER can be extracted from mapping curves as a function of the Signal-to-Noise Ratio (SNR) of the MRT link [13] or the FSO link [29]. With respect to the SNR of an MRT link, it can be given by (in dB):

$$\text{SNR}^{(\text{MRT})} = P_{Rx} - N_f - N_0, \tag{19}$$

where $N_f$ corresponds to the noise figure of the receiver and $N_0$ denotes the thermal noise, which can be computed as (in dBW):

$$N_0 = -204 + 10 \log_{10}(b_{w[\text{Hz}]}), \tag{20}$$

where $b_w$ represents the noise equivalent bandwidth of the receiver; it can be expressed as:

$$b_w = \frac{B_{link}}{\log_2(M)}, \tag{21}$$

where $B_{link}$ and $M$ stand for the link bit rate and the QAM (Quadrature Amplitude Modulation) signal constellation size, respectively. Considering an FSO link and making the typical assumption of a shot-noise-limited operation with On-Off Key (OOK) modulation, the SNR can be obtained as (in dB):

$$\text{SNR}^{(\text{FSO})} = P_{Rx} - \frac{P_{Rx} + A_{turb}}{2} - 5 \log_{10} \left( 2 \, h \times \frac{c}{\lambda} \times B_{link} \right), \tag{22}$$

where $h$ and $c$ refer to the Planck constant and the speed of light, respectively.

Considering now FO systems, the associated technology differs from the two wireless communication systems previously discussed (MRT and FSO) as it does not use the atmosphere as the propagation medium. More specifically, since the beam is confined to the fiber, there are no outside weather conditions that need to be taken into consideration when planning point-to-point transmission using the FO technology. Accordingly, evaluating a link budget for FO is equivalent to computing the total loss, suffered by a transmitted signal across various components and along the optical fiber, with reference to the minimum receiver power required to maintain normal operation.

In mathematical terms [30], the FO link budget, $L_B$, is given by (in dB):

$$L_B = Tx_{min} - Rx_{min}, \tag{23}$$

where $Tx_{min}$ and $Rx_{min}$ correspond to the minimum transmit power (at the transmitter) and minimum received power required (at the receiver), respectively (both in dBW or dBm). The total loss suffered by the transmitted signal along the link, $T_L$, is given by (in dB):

$$T_L = L + (d \times F_L), \tag{24}$$

where $L$ stands for the losses in optical connectors (in dB) and $F_L$ corresponds to the normalized fiber loss (in dB per units of distance); typical values for these parameters can be found in the related literature [30]. Finally, an FO link is assumed to be feasible if the FO link margin, i.e., $L_B - T_L$, is greater than a user-specified minimum accepted link margin (e.g., 3 dB, as adopted in this work), and if the link bit rate $\times$ distance product (i.e., the required bit rate times the link length) does not exceed the maximum bit rate $\times$ distance product of the fiber [10].

### 3.1.2. Link Costs Analysis

When planning a link, the costs associated with the project will always go beyond the costs of the equipment itself; it is important to mention that when referring to the equipment of a certain technology (MRT, FSO, or FO), it includes all the necessary items for installing and operating the respective link. In particular, two different types of costs have to be considered in the scope of these projects:

- CAPEX: These include the fixed costs related to the network infrastructure, such as equipment and respective deployment, spare parts, and project studies. With respect to wireless technologies, one has to consider emitter and receiver costs, as well as the costs of cables, stands, and in the case of FSO systems, an auto-tracker that allows the receptor to align with the received signal (in order to reduce the impact of atmospheric turbulence). Moreover, as stated in Section 2, the costs associated with CAPEX for MRT and FSO technologies do not depend on the link distance (if no repeaters are considered), unlike FO links. Accordingly, CAPEX related to FO systems can be divided into a fixed term, which accounts for Optical Line Terminations (OLTs), Optical Network Units (ONUs), and other miscellaneous electronics, plus a variable term that corresponds to the costs of the fiber itself and the costs of deploying it, which varies with the length of the link, $d$.
- OPEX: These do not contribute to the infrastructure itself, since they include operational expenses, such as maintenance costs, energy consumption, government taxes, and repayments. Accordingly, any economical analysis regarding OPEX is usually performed taking into account the lifetime of the communication link—e.g., a period of 10 years, as adopted in this work.

After gathering the values for both CAPEX and OPEX$_{\text{lifetime}}$ (for a given lifetime) regarding the use of a certain equipment, the total costs of the link project are given by the summation of these values, i.e.,

$$\text{Total Costs} = \text{CAPEX} + \text{OPEX}_{\text{lifetime}}. \tag{25}$$

Based on what was previously mentioned and recalling the information provided in Section 2, the total costs depend on the link length in the case of MRT and FO systems, whereas the total costs can be regarded as a fixed value (i.e., independent of the link length) for FSO systems; thus, Expression (25) can be rewritten with respect to the different technologies as:

$$\text{Total Costs}^{(\text{MRT})} = \text{F.Costs}^{(\text{MRT})} + \text{V.Costs}^{(\text{MRT})} \times \sqrt{d}, \tag{26}$$

$$\text{Total Costs}^{(\text{FSO})} = \text{F.Costs}^{(\text{FSO})}, \tag{27}$$

$$\text{Total Costs}^{(\text{FO})} = \text{F.Costs}^{(\text{FO})} + \text{V.Costs}^{(\text{FO})} \times d, \tag{28}$$

where F.Costs and V.Costs stand for fixed costs and variable costs, respectively, with respect to the associated technology (MRT, FSO, or FO); please note that these costs (i.e., F.Costs$^{(\text{MRT})}$, F.Costs$^{(\text{FSO})}$, F.Costs$^{(\text{FO})}$, V.Costs$^{(\text{MRT})}$, and V.Costs$^{(\text{FO})}$) also represent user-specified inputs of the link design algorithm.

### 3.1.3. Link Design Algorithm Workflow

Figure 2 depicts the flowchart of the link design algorithm. The first step is to read and store the information contained in the "MRT.dat", "FSO.dat", and "FO.dat" files. These ".dat" files are text files that contain data about the user-specified equipment being tested, as well as about the respective

associated costs; in general, these necessary inputs are provided by the equipment manufacturers and, concerning costs, by taking into account the CAPEX and OPEX items listed in the previous section. The user can test and compare, at the same time, as many different equipment as desired, as the algorithm is able to process a variable amount of equipment; in this manner, the user can, for example, test and compare different solutions from different providers in a single run of the algorithm. More specifically, each line of the ".dat" file associated with the respective communication technology (MRT, FSO, and FO) corresponds to a different equipment of that technology. The structure of each line of these ".dat" files, namely the required inputs for each communication technology equipment (which should be separated by commas), is given in Table 2, where ID refers to the identifier (number or word) of an equipment, $B$ stands for the maximum bit rate that an equipment can offer for the specified inputs, whereas $BxD$ corresponds to the maximum bit rate $\times$ distance product of an optical fiber.

**Table 2.** Required inputs for the "MRT.dat", "FSO.dat", and "FO.dat" files.

| MRT .dat | ID | $B$ (Mbps) | $f$ (GHz) | $P_{Tx}$ (dBW) | $G_{Tx}$ (dBi) | $G_{Rx}$ (dBi) | $A_{equi}$ (dB) | $S_{Rx}$ (dBW) | $N_f$ (dB) | $M$ ($M$-QAM) | F.Costs$^{(MRT)}$ (¤) | V.Costs$^{(MRT)}$ (¤/$\sqrt{km}$) |
|---|---|---|---|---|---|---|---|---|---|---|---|---|
| FSO .dat | ID | $B$ (Mbps) | $\lambda$ (nm) | $P_{Tx}$ (dBW) | $G_{Tx}$ (dBi) | $G_{Rx}$ (dBi) | $A_{equi}$ (dB) | $S_{Rx}$ (dBW) | F.Costs$^{(FSO)}$ (¤) | | | |
| FO .dat | ID | $B$ (Mbps) | $BxD$ (Mbps·km) | $Tx_{min}$ (dBW) | $Rx_{min}$ (dBW) | $L$ (dB) | $F_L$ (dB/km) | F.Costs$^{(FO)}$ (¤) | | V.Costs$^{(FO)}$ (¤/km) | | |

In addition, the user must provide data (such as link length, required bit rate, and climate information) regarding the link deployment scenario. This information is given in the "Scenario.dat" text file and follows a line structure (separated by commas), where the required inputs are indicated in Table 3; these correspond respectively to link length ($d$), required bit rate ($B_{min}$), maximum percentage of the time regarding tolerated link unavailability ($U_{max}$), temperature ($T$), rain intensity not exceeded in 0.01% of the time ($R_i^{(0.01\%)}$), relative humidity ($H$), transmitter altitude ($h_a$), height difference between the top of an obstacle and the line-of-sight ($h_{obs}$), average number of foggy days ($\overline{N_{fog}}$), average duration of fog events ($\overline{D}$), and minimum accepted link margins for MRT, FSO, and FO links ($Ml_{min}^{(MRT)}$, $Ml_{min}^{(FSO)}$, and $Ml_{min}^{(FO)}$, respectively).

From here, the algorithm becomes independent of the user. It will go through all the $N$ different user-specified equipment, and for each one, the algorithm first checks if the required bit rate of the link is met by the equipment; if that condition is satisfied, then it is evaluated if the link is feasible with that equipment (taking into account the communication technologies models); in other words, if the required link distance is shown to be too long to be accommodated in a single-hop by the equipment that is under consideration in each iteration, then that equipment is ignored in the remaining analysis. Finally, the algorithm returns the cheapest working solution (taking into account the link costs analysis), namely the respective equipment ID and total costs. Please note that if none of the $N$ different user-specified pieces of equipment meet the link requirements, then the algorithm returns (positive) infinity for the total costs and no ID.

**Table 3.** Required inputs for the "Scenario.dat" file.

| Scenario.dat | $d$ (km) | $B_{min}$ (Mbps) | $U_{max}$ (%) | $T$ (°C) | $R_i^{(0.01\%)}$ (mm/h) | $H$ (%) | $h_a$ (m) | $h_{obs}$ (m) | $\overline{N_{fog}}$ (days/year) | $\overline{D}$ (h) | $Ml_{min}^{(MRT)}$ (dB) | $Ml_{min}^{(FSO)}$ (dB) | $Ml_{min}^{(FO)}$ (dB) |
|---|---|---|---|---|---|---|---|---|---|---|---|---|---|

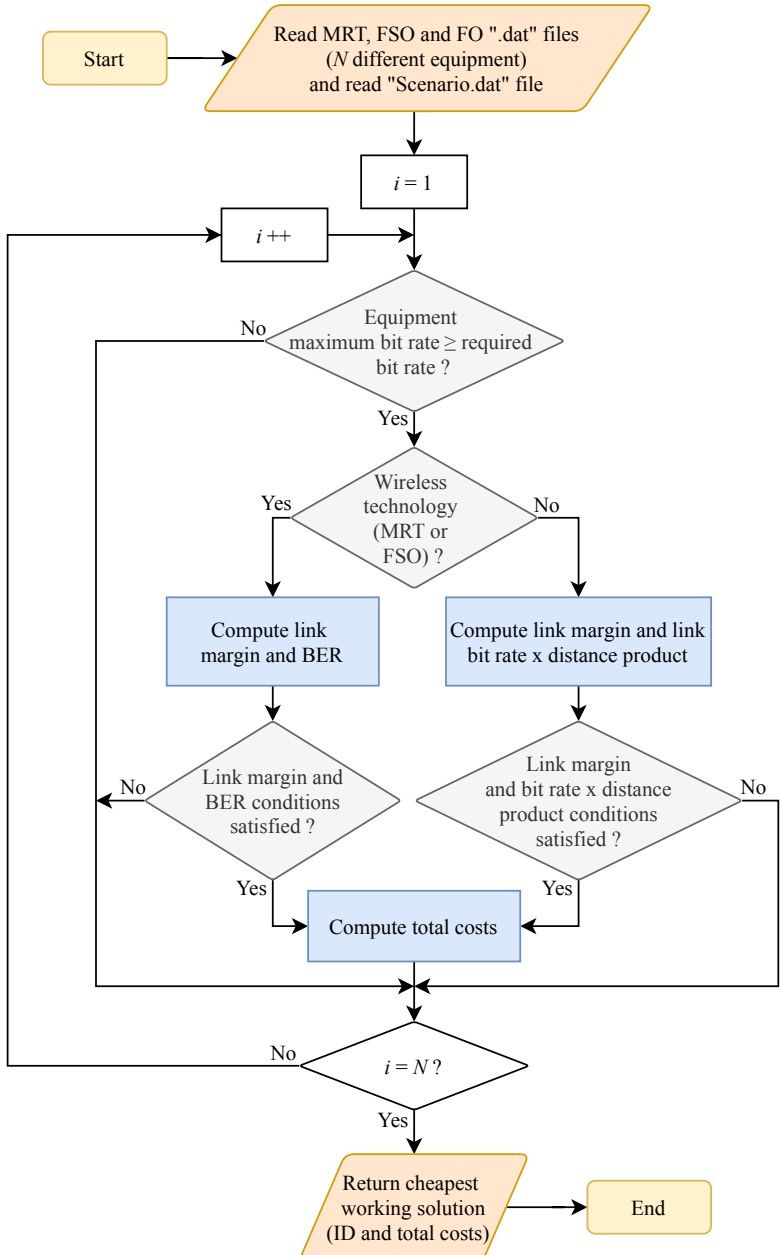

**Figure 2.** Flowchart of the link design algorithm.

### 3.2. Network Planning Algorithm

The network planning algorithm has the goal of finding the optimal number of BBUs to be deployed for a certain environment and where to place them, given the positions of the RRHs and their bit rate needs. Using the previously described link design algorithm and considering the costs associated with a BBU, this algorithm is able to compute fronthaul topologies that minimize the total costs of the network.

The following subsection details the network planning algorithm, namely the considered economic aspects and the algorithm workflow.

### 3.2.1. Network Costs Analysis

With respect to the total costs of the network, there are two main aspects to take into consideration: the costs of each RRH–BBU link and the costs of a BBU. Accordingly, the network total costs are given

by the sum of the global costs of the RRH–BBU links plus the costs of the total number of BBUs used in the network.

The costs of each RRH–BBU link can be obtained with the link design algorithm. It is important to notice that when dealing with the acquisition of equipment, as is the case when designing the fronthaul network, some vendors might be able to provide substantial discounts on their prices for purchases involving larger amounts of equipment. Accordingly, these discounts could make one technology preferable to another, cost-wise. Nevertheless, even though this is an important consideration, it is extremely difficult to quantify these discounts as constraints within the algorithm in a universal manner. Therefore, it was assumed that the unitary price of the equipment remained unaltered for large quantities, which means that the global costs of the RRH–BBU links considered by the network planning algorithm can be regarded as an upper limit for this type of cost; nonetheless, please bear in mind that since the source code of this algorithm is freely available online, it can be modified by anyone, namely by a system designer, in order to incorporate other features such as particular costs' computations according to specific terms of the vendors.

It is worth pointing out that when deciding to add or remove a BBU from the network, the BBU costs play a major role when determining the total costs of the network. More specifically, if the BBU costs were considered negligible, the optimal solution (although unrealistic) would be given by the total number of BBUs equaling the number of RRHs, with colocated placements, as this would mean close to zero distances between the RRHs and the corresponding BBU, thus yielding the minimum possible costs regarding RRH–BBU links. Accordingly, one of the virtues of the network planning algorithm is that the BBU costs are not disregarded when determining the optimal number of BBUs to be deployed for a certain environment.

### 3.2.2. Network Planning Algorithm Workflow

Figure 3 depicts the flowchart of the network planning algorithm. The first step is to read and store the information contained in the "RRH.dat" file, which is a text file that contains (in each line and separated by commas) the Cartesian coordinates $(X, Y)$ of the RRHs under consideration, along with the required bit rate for each RRH; Table 4 presents the structure of each line regarding the "RRH.dat" file. In addition, the user must provide data regarding the BBUs in the "BBU.dat" text file; once more, a line structure is followed (separated by commas), where the required inputs are the ones indicated in Table 5; these correspond respectively to the maximum number of RRHs supported by a BBU ($RRHs_{max}$), the maximum bit rate supported by a BBU for each RRH–BBU link ($B_{max}$), the costs of a BBU ($Costs_{BBU}$), the minimum and maximum number of BBUs to be considered in the analysis (*min_BBU* and *max_BBU*, respectively), and the number of different initializations regarding the network costs optimization procedure ($D_{init}$).

Afterwards and given the starting number of BBUs (*min_BBU*), the algorithm computes the positions of the BBUs. This step, combined with the assignment of each RRH to a BBU (where multiple RRHs can be connected to a single BBU), can be regarded as solving a clustering problem. Since smaller RRH–BBU link distances not only increase the likelihood of adopting single-hop wireless communication technologies (which are simpler to install), but also lead to cheaper FO links, as well as lower fronthaul latencies can be achieved, the adopted criterion for clustering is based on minimizing the distances between RRHs and BBUs. Accordingly and by applying the *K*-means clustering algorithm (as suggested in [31]), the positions of the BBUs are determined. More specifically, the *K*-means procedure groups the RRHs into *K* different subsets, where *K* equals the considered number of BBUs (hence, $K \leq RRHs$), by minimizing the sum of squared distances between the RRHs belonging to a cluster and the corresponding cluster centroid, i.e., the associated BBU position.

**Table 4.** Required inputs for the "RRH.dat" file.

| **RRH.dat** | $X$ (m) | $Y$ (m) | $B_{min}$ (Mbps) |
|---|---|---|---|

**Table 5.** Required inputs for the "BBU.dat" file.

| BBU.dat | $RRHs_{max}$ | $B_{max}$ (Mbps) | $Costs_{BBU}$ (¤) | min_BBU | max_BBU | $D_{init}$ |
|---|---|---|---|---|---|---|

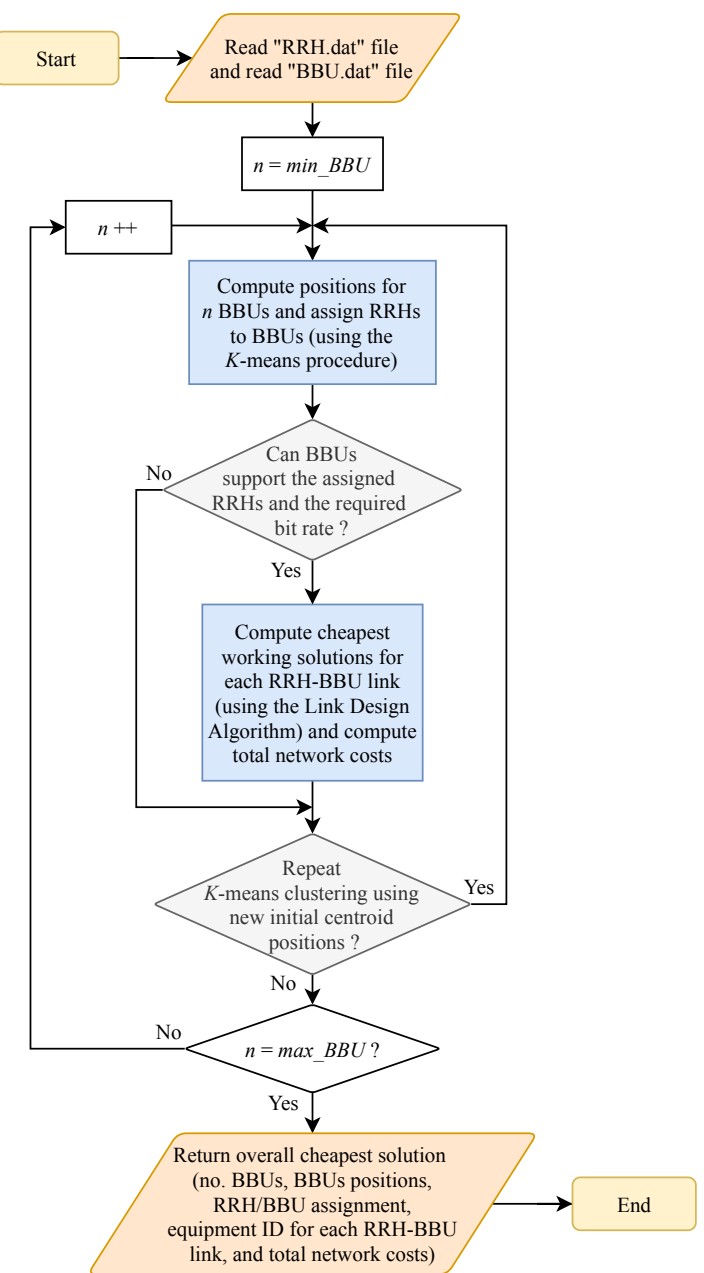

**Figure 3.** Flowchart of the network planning algorithm.

Next, the algorithm verifies if the BBUs can support the number of RRHs assigned to each one of them, as well as the required bit rate of each RRH–BBU link; if this is the case, then the link design algorithm is used to determine the cheapest working solution satisfying each RRH–BBU link requirement (such as link distance and required bit rate) for the deployment scenario (the scenario characteristics, except link distance and required bit rate, are extracted from the "Scenario.dat" file); cf. Section 3.1. Subsequently, the total costs of the network are computed taking into account the network costs analysis, i.e., by summing up the costs of all RRH–BBU links and the costs of the considered number of BBUs.

Noticing that the *K*-means approach is an iterative algorithm that relies on an initial random choice of centroid positions, this procedure may yield different clustering results on different runs of the algorithm, which, in turn, may lead to different network costs; note that there are $K^{N(RRHs)}$ ways to partition *N* RRHs into *K* clusters. Therefore, the previous two steps (computation of BBUs' positions and determination of the total costs of the network) are repeated multiple times—namely a total of $D_{init}$ times, which is a user-specified value that ideally should be close to $K^{N(RRHs)}$, although this number may be too time consuming to be practical—in order to search for the cheapest network topology regarding the number of BBUs that is under consideration (e.g., *min_BBU*).

The network planning algorithm then computes again the total costs of the network, but now considering a different number of BBUs, in order to find the cheapest alternative. More specifically, the procedure is repeated, sequentially, from the minimum (*min_BBU*) to the maximum number of BBUs (*max_BBU*) defined by the user, and the algorithm will finally return which number of BBUs yields the overall cheapest solution along with other outputs: BBUs positions, RRHs assigned to each BBU, equipment ID for each RRH–BBU link, and total network costs.

## 4. Fronthaul Design Study Results

In order to study the influence of different environments and weather conditions on the fronthaul design, a set of illustrative equipment was considered: it comprised MRT, FSO, and FO technologies under different functioning conditions, such as maximum supported bit rate, price, and others factors that affected the maximum distance of a viable link (e.g., operating frequency, sensitivity, etc.). The respective test equipment characteristics, which were obtained by combining information from multiple sources [15,32–35], are presented in Table 6.

**Table 6.** Characteristics of the test equipment.

| MRT | ID | $B$ (Mbps) | $f$ (GHz) | $P_{Tx}$ (dBW) | $G_{Tx}$ (dBi) | $G_{Rx}$ (dBi) | $A_{equi}$ (dB) | $S_{Rx}$ (dBW) | $N_f$ (dB) | $M$ ($M$-QAM) | F.Costs$^{(MRT)}$ (EUR) | V.Costs$^{(MRT)}$ (EUR/$\sqrt{km}$) |
|---|---|---|---|---|---|---|---|---|---|---|---|---|
| | 1 | 380 | 13 | 2 | 28 | 28 | 3 | −111 | 4 | 1024 | 55,895 | 11,590 |
| | 2 | 570 | 42 | 0 | 34 | 34 | 3 | −122 | 4 | 1024 | 85,326 | 5016 |
| | 3 | 1100 | 42 | 0 | 34 | 34 | 3 | −122 | 4 | 1024 | 95,736 | 9680 |
| **FSO** | ID | $B$ (Mbps) | $\lambda$ (nm) | $P_{Tx}$ (dBW) | $G_{Tx}$ (dBi) | $G_{Rx}$ (dBi) | $A_{equi}$ (dB) | $S_{Rx}$ (dBW) | F.Costs$^{(FSO)}$ (EUR) | | | |
| | 4 | 100 | 785 | −12.2 | 30 | 0 | 3 | −60 | 29,903 | | | |
| | 5 | 1000 | 1550 | −27 | 30 | 0 | 3 | −60 | 32,353 | | | |
| | 6 | 1000 | 1550 | −7 | 30 | 0 | 3 | −60 | 48,186 | | | |
| | 7 | 10,000 | 1550 | −7 | 30 | 0 | 3 | −48 | 86,654 | | | |
| **FO** | ID | $B$ (Mbps) | BxD (Mbps·km) | $Tx_{min}$ (dBW) | $Rx_{min}$ (dBW) | $L$ (dB) | $F_L$ (dB/km) | F.Costs$^{(FO)}$ (EUR) | | V.Costs$^{(FO)}$ (EUR/km) | | |
| | 8 | 10,000 | 80,000 | −27.5 | −43 | 3 | 0.25 | 35,135 | | 18,865 | | |

The study performed herein is divided into two parts: first, the link design algorithm is used to evaluate which technologies are the preferable cost-effective choices for individual fronthaul segments under different weather conditions, lengths of the links, and bit rate requirements; secondly, by using the network planning algorithm, an assessment is performed regarding the impact of the RRH density on the selection of cost-effective communication technologies for C-RANs.

### 4.1. Individual Links Assessment

In order to study the performance of MRT, FSO, and FO systems, four different scenarios were considered by taking into account the ranges of values associated with rain and fog intensities [25,28]; these test scenarios are henceforth denoted as S(r_l/f_l), S(r_h/f_l), S(r_l/f_h), and S(r_h/f_h) and their respective characteristics are given in Table 7; for the sake of clarity, r_l and r_h refer to low and high rain

intensity, respectively, whereas f_l and f_h refer to low and high fog intensity, respectively. In addition, the minimum accepted link margins $Ml_{min}^{(MRT)}$, $Ml_{min}^{(FSO)}$, and $Ml_{min}^{(FO)}$ were all set equal to 3 dB [10,32].

**Table 7.** Characteristics of the test scenarios.

| Scenario | $U_{max}$ (%) | $T$ (°C) | $R_i^{(0.01\%)}$ (mm/h) | $H$ (%) | $h_a$ (m) | $h_{obs}$ (m) | $\overline{N_{fog}}$ (days/year) | $\overline{D}$ (h) | Rain Intensity | Fog Intensity |
|---|---|---|---|---|---|---|---|---|---|---|
| S(r_l/f_l) | | | 15 | | | | 10 | | Low | Low |
| S(r_h/f_l) | | | 135 | | | | 10 | | High | Low |
| S(r_l/f_h) | 0.1 | 21 | 15 | 60 | 30 | −15 | 360 | 2 | Low | High |
| S(r_h/f_h) | | | 135 | | | | 360 | | High | High |

Following the previous test scenario settings, the link design algorithm was run for each scenario considering a set of distances ranging from 0 to 20 km and a required bit rate ranging from 0 to 1.5 Gbps, in order to assess how the most cost-effective solution varied for different link requirements and for different weather conditions. Figures 4–7 present the outputs of the link design algorithm (i.e., the cheapest working solution for the considered distance and required bit rate ranges) for the four test scenarios.

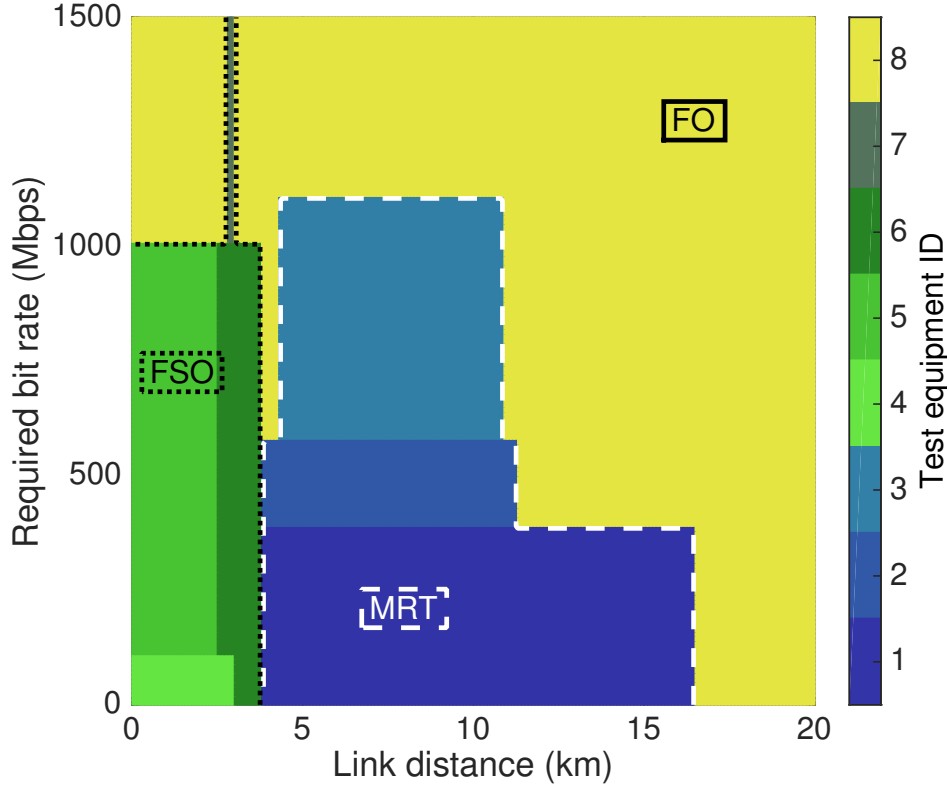

**Figure 4.** Cost-effective solutions for S(r_l/f_l).

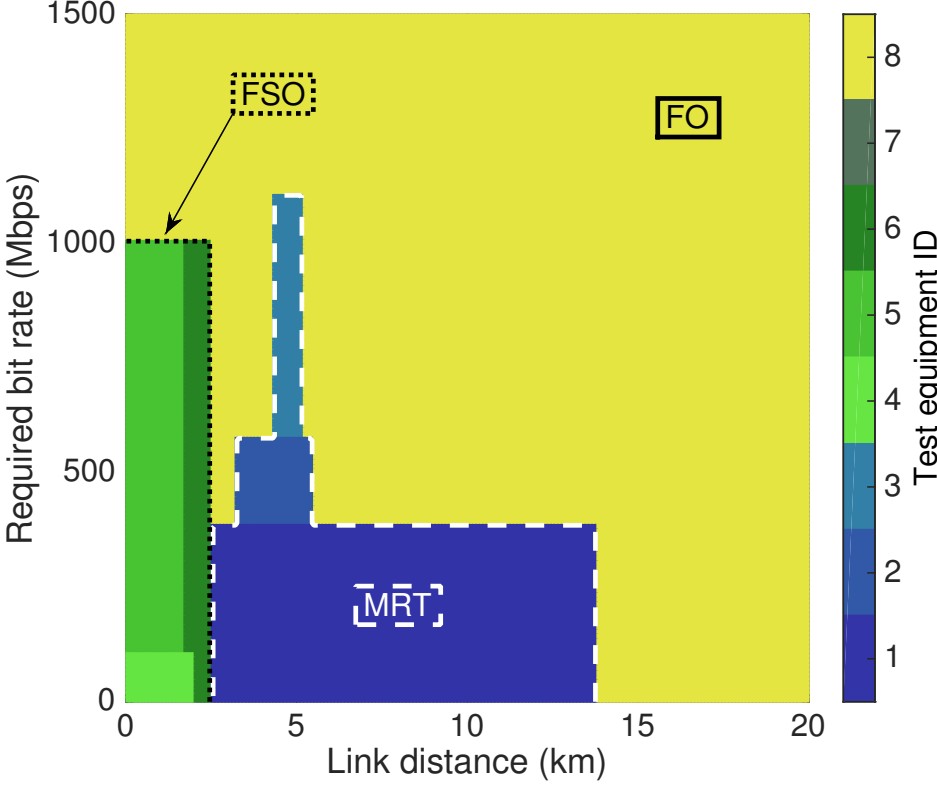

**Figure 5.** Cost-effective solutions for S(r_h/f_l).

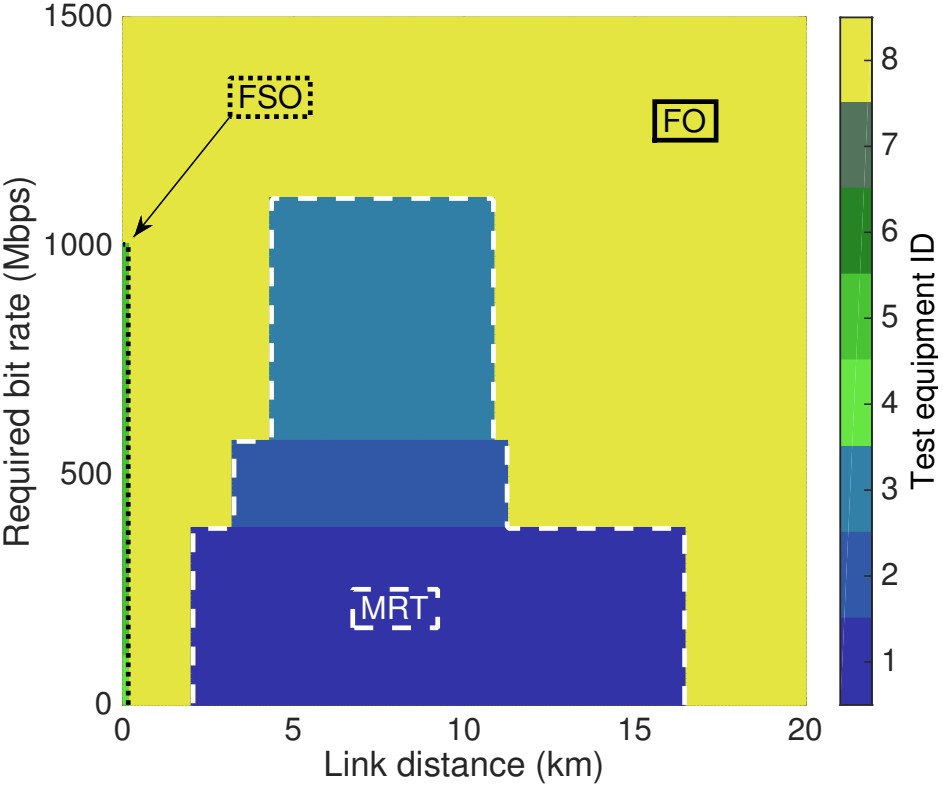

**Figure 6.** Cost-effective solutions for S(r_l/f_h).

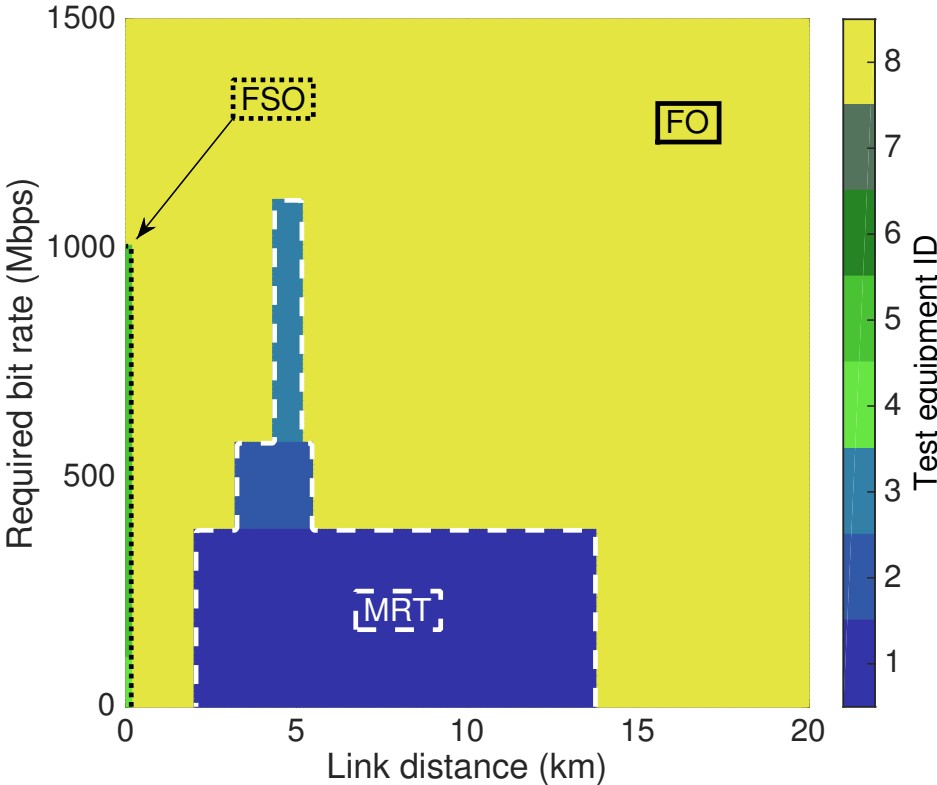

**Figure 7.** Cost-effective solutions for S(r_h/f_h).

Due to the low rain and low fog intensities considered in S(r_l/f_l), this scenario has very good wireless propagation conditions for both MRT and FSO technologies. Accordingly and as can be observed in Figure 4, an FSO system should be adopted for these weather conditions when the required bit rate is low to medium (lower than 1 Gbps) and the link distance is small (below 4 km) or when the required bit rate is higher (1 to 1.5 Gbps, at least) and the link distance is within a narrow range of small distances (2.5 to 3 km); for the medium bit rate requirement range (570 Mbps to 1.1 Gbps) and for medium link distances (4.5 to 11 km), MRT systems are the best choice; the MRT technology also yields the best results for a wider range of link distances (namely from 4 to 11.5 km and from 4 to 16.5 km), but for lower bit rate requirements (below 570 Mbps and below 400 Mbps, respectively); for all the remaining cases, FO systems are the best choice. Some conclusions can be drawn from these results: first, there is a trade-off between bit rate and distance ranges regarding the cost-effectiveness of MRT and FSO solutions; secondly, the feasibility of a wireless link does not necessarily mean that it is a cost-effective solution; for instance, although FSO links are practicable regarding small distances and high bit rate requirements (notably greater than 1 Gbps), FO systems are a better cost-effective choice in these cases if the link distance is closer to zero (an equivalent example could be given regarding MRT versus FO systems and with respect to small to medium link distances and medium bit rate requirements); nevertheless, FSO solutions can be used as a fallback in these cases, namely if the FO technology cannot be used, e.g., due to the orography of the terrain or the necessity of a quick installation following a natural disaster.

Considering now S(r_h/f_l), which is a scenario characterized by high rain and low fog intensities, Figure 5 shows that both wireless technologies decreased their cost-effective performance in terms of maximum link distance when compared to S(r_l/f_l); this performance decrease was more pronounced regarding MRT systems, especially in the case of medium bit rate requirements (they had a reduction of 6 km regarding the maximum feasible link length). From here, it could be concluded that although the presence of rain makes it harder to select one of the wireless technologies considered herein, if the fog intensity is low, then FSO systems are still a good cost-effective option if the required bit rate is low

to medium (lower than 1 Gbps) and the link distance is small (below 2.5 km); on the other hand, even in the case of high rain intensity, MRT solutions should not be disregarded, since they are cost-effective for medium link distances (2.5 to 14 km) and small bit rate requirements (lower than 400 Mbps).

Turning now the attention to the scenario characterized by low rain and high fog intensities, S(r_l/f_h), Figure 6 shows that FSO systems were practically excluded from the cost-effective solutions due to harsh visibility conditions; in particular, FSO links were only feasible for very small distances (up to 200 m). With respect to MRT systems, their cost-effectiveness was similar to the one observed for S(r_l/f_l) (cf. Figure 4). Another noteworthy result is that, when compared to the previously analyzed scenarios, it was the FO technology (and not the MRT one) that replaced FSO as the cost-effective solution for small link distances (below 2 km) and low to medium bit rate requirements (lower than 1 Gbps); this result reinforces the previous conclusion that the feasibility of a wireless link does not necessarily mean that it is a cost-effective solution, namely when considering small link distances.

With respect to S(r_h/f_h), which is a scenario characterized by high rain and high fog intensities, Figure 7 shows that the cost-effectiveness of MRT systems was similar to the one observed for S(r_h/f_l) (cf. Figure 5), whereas the cost-effectiveness of FSO systems was similar to the one observed for S(r_l/f_h) (cf. Figure 6), thus yielding analogous conclusions, respectively.

Combining all the findings gathered so far, a system designer, when planning an RRH–BBU link, can somewhat foresee, without making computations, which technology is more likely to be adopted; this is useful to decide, especially in the first stage of link planning, which equipment should be thoroughly surveyed on the market, thus saving time and money. A summary of the mapping between the pair link distance/bit rate requirement and the respective cost-effective solutions is given in Figure 8, which led to the following rules of thumb:

- For short link distances (under 4 km) and small to medium bit rate requirements (lower than 1 Gbps), an FSO system is the best option, followed by an FO one if the former is unfeasible, notably in foggy scenarios (MRT systems can also be used as fallback);
- For short link distances (under 3 km), but for a higher bit rate requirement (greater than 1 Gbps), FO systems should be adopted, followed by FSO ones;
- For medium to high link distances (5 to 11 km) and small to medium bit rate requirements, as well as some high link distances (14 to 16 km) and lower bit rate requirements (lower than 400 Mbps), the first choice corresponds to the MRT technology, followed by FO if the former is unfeasible, notably in rainy scenarios;
- For medium to high link distances (4 to 14 km) and lower bit rate requirements, as well as some medium link distances (4 to 5 km) and medium bit rate requirements (lower than 1 Gbps), not only MRT systems are the best option, but they also show a high resilience even regarding scenarios with high rain intensity;
- For all the remaining cases, FO systems are the only option, which means that if these systems cannot be deployed, then the link becomes unfeasible (unless wireless relay stations are considered).

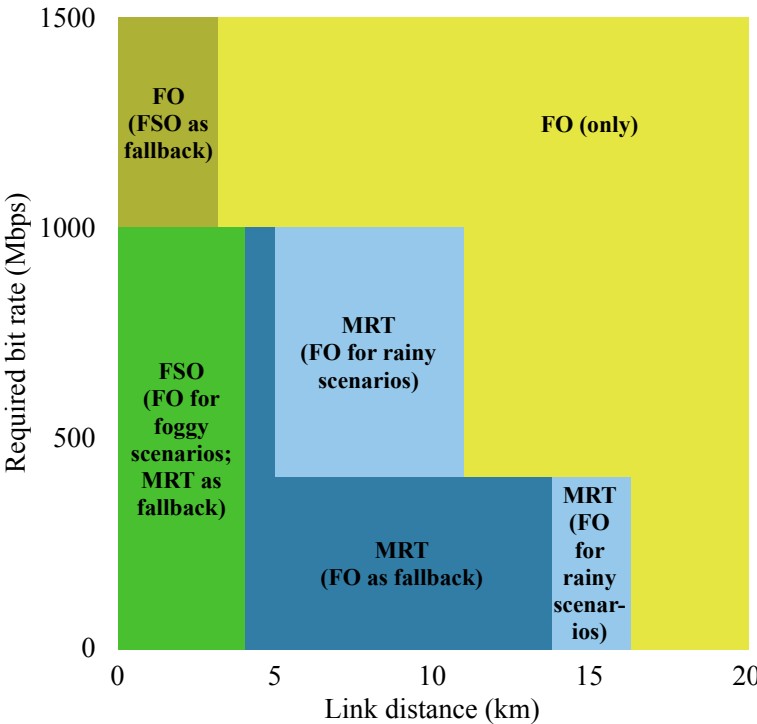

**Figure 8.** Summary of the cost-effective solutions for individual link deployment.

*4.2. Network Assessment*

In order to assess the impact of the RRH density on the selection of cost-effective communication technologies for C-RANs, two distinct environments were considered, namely a rural and an urban environment [36], which had different RRH densities; the characteristics of these test environments are presented in Table 8. In addition, the costs of one BBU was assumed to be equal to 167,000 euros [33,37], as well as it was assumed that each BBU had enough capacity to support as many RRH–BBU links as needed; furthermore, for each of these links, it was assumed that the BBUs could support the required bit rate, which was set equal to 1 Gbps for all cases. Moreover, the *K*-means clustering procedure was repeated enough times so that each solution using a certain number of BBUs converged to a minimum in terms of network costs.

**Table 8.** Characteristics of the test environments for network assessment.

| Environment | RRH Density | Test Area | Total No. of RRHs |
|---|---|---|---|
| Rural | 0.015 RRHs/km$^2$ | 20 km $\times$ 20 km | 6 |
| Urban | 0.3 RRHs/km$^2$ | | 120 |

Following the previous settings, the network planning algorithm was run multiple times for each environment and considering, in a separate manner, the four aforementioned test scenarios (cf. Table 7), in order to evaluate the impact of different weather conditions on the fronthaul design and costs.

4.2.1. Rural Environment

Figure 9 presents, for each of the considered test scenarios, the obtained results for the rural environment, namely the average total network costs along with the average technology usage percentage; the colocated percentage corresponds to the average percentage of RRHs that were colocated with the placed BBUs; thus, the respective RRH–BBU links did not make use of MRT, FSO, or FO individual links. Additionally, the average number of BBUs required for each test scenario is also indicated in this figure.

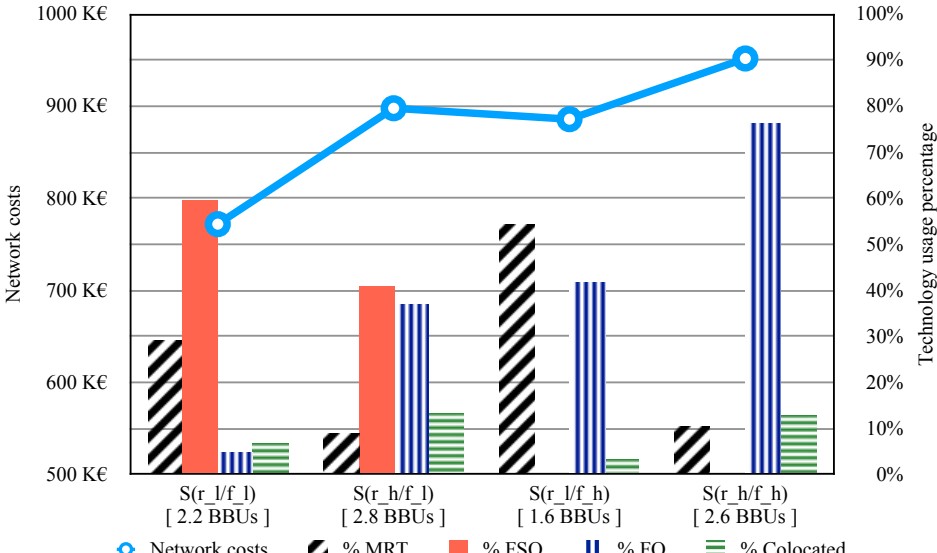

**Figure 9.** Network costs and technology usage for the rural environment.

The results showed that even though the average number of BBUs required for a cost-effective network design was similar in all cases (ranging from 1.6 to 2.8), the distribution of technology usage regarding the RRH–BBU links clearly varied according to the considered scenario. Nevertheless, it is important to notice that, in all cases, cost-effective solutions were obtained (on average) when some BBUs were colocated with RRHs. Moreover, the FO technology was always adopted for RRH–BBUs links, and its usage percentage increased as the weather conditions deteriorated.

As expected, the scenario characterized by low rain and low fog intensities, S(r_l/f_l), yielded the lowest average network costs, whereas the high rain and high fog intensities of S(r_h/f_h) led to the highest average network costs. However, these scenarios did not give rise to the lowest or highest average number of required BBUs, respectively, concerning all four test scenarios. This shows that the number of required BBUs on its own is not sufficient to have an estimate regarding the final network costs in rural environments.

Another noteworthy result is that regardless of the rain intensity, MRT systems were always adopted for RRH–BBUs links, albeit with lower usage percentages when the rain intensity was high. On the contrary, an equivalent behavior was not observed by the FSO technology; more specifically, although this technology was the most used when the fog intensity was low, FSO systems were not adopted in the scenarios with high fog intensity. This particular outcome in the rural environment enables to highlight an advantage of considering the FSO technology when planning fronthaul networks: if the visibility conditions are favorable, then the use of FSO systems entails significant savings in the network cost; for instance, there was a cost reduction of 6% in rainy scenarios (cf. S(r_h/f_h) vs. S(r_h/f_l)), whereas a cost reduction of 13% was verified under non-rainy scenarios (cf. S(r_l/f_h) vs. S(r_l/f_l)). Furthermore and besides the cost savings, the use of FSO systems leads to an increase of the number of BBUs required for a cost-effective network, which may represent an advantage in future network expansions, especially in rural environments, as new RRHs have a higher likelihood of being served with lower costs by the existing BBUs.

4.2.2. Urban Environment

Figure 10 presents, for each of the considered test scenarios, the obtained results for the urban environment. As can be observed, the distribution of technology usage regarding the RRH–BBU links can be categorized into two groups, according to the visibility conditions: (1) low fog intensity scenarios, where the FSO technology usage is largely dominant (roughly 90–98%) and FO systems account for the remaining RRH–BBU links, with a usage percentage that increases along with the increase of the rain

intensity (particularly because the performance of FSO systems is also impaired by the presence of rain, as seen in Section 4.1); and (2) high fog intensity scenarios, in which the distribution of technology usage is practically unaffected by the rain intensity and almost all RRH–BBUs links (about 99%) are served by FO systems. Still regarding the latter group, it is important to notice that even in the presence of harsh visibility conditions in the respective scenarios, FSO systems were adopted for the remaining RRH–BBU links (i.e., approximately for 1% of the cases); this occurred because, unlike the rural environment, some of the RRH–BBU link lengths were always very small (notably less than 200 m) due to the higher RRH density of the urban environment.

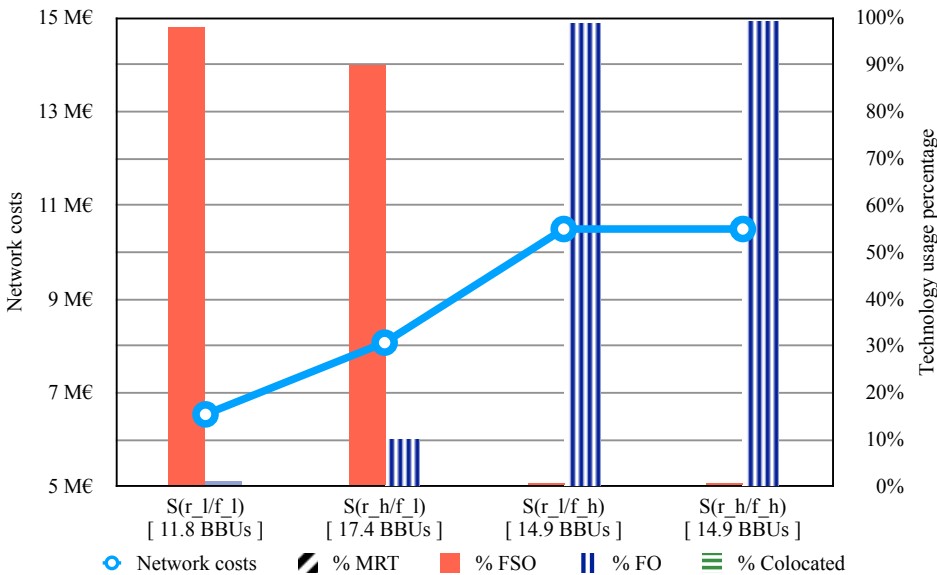

**Figure 10.** Network costs and technology usage for the urban environment.

Turning the attention now to the network costs, the results showed that the visibility conditions also have a major impact on the urban fronthaul costs: not only high fog intensities led to more expensive networks, but also cheaper ones were obtained when there was low fog intensity; in particular, the possibility of exploiting FSO systems when the visibility conditions were favorable led to a cost reduction of 23% in rainy scenarios (cf. S(r_h/f_h) vs. S(r_h/f_l)), whereas a cost reduction of 38% was verified under non-rainy scenarios (cf. S(r_l/f_h) vs. S(r_l/f_l)). It is also worth to point out that, regardless of the rain intensity, the network costs were very similar whenever the fog intensity was high (cf. S(r_l/f_h) vs. S(r_h/f_h)), mainly because MRT systems were not adopted; in any case, although it is unnoticeable in Figure 10, the costs were slightly cheaper in S(r_h/f_l) than in S(r_l/f_l)) owing to a non-negligible performance increase of FSO systems when the rain intensity decreased, which in turn slightly increased the FSO technology usage.

With respect to the average number of required BBUs, although the lowest value was obtained for the scenario that yielded the lowest average network costs (i.e., S(r_l/f_l)), the opposite was not true, as the highest number of required BBUs occurred for the second cheapest scenario (i.e., S(r_h/f_l)). This shows that the number of required BBUs on its own is not sufficient to have an estimate regarding the final network costs also in urban environments. Finally, notice that although colocated BBUs with RRHs were never adopted as cost-effective solutions regarding the urban environment results, in practice, it may compensate economical to change the placement of a BBU (which was initially suggested by preliminary cost-effectiveness studies) in order for it to match the position of a nearby RRH (for instance, if the initial distance between them is below 20 m), thus saving, e.g., site rental costs.

## 5. Conclusions

This paper addressed the use of MRT, FSO, and FO systems in the C-RAN fronthaul segment of a mobile network. In particular, a methodology was proposed to evaluate and compare the performance of MRT, FSO, and FO technologies, thus enabling determining the most cost-effective solution for each RRH–BBU link and computing the required number of BBUs and where they should be positioned in order to minimize the overall network costs. In addition, a study was carried out regarding the fronthaul design; more specifically, a cost-effectiveness comparison of the aforementioned communication technologies was performed for individual fronthaul segments under different weather conditions, link lengths, and bit rate requirements; an assessment was also performed regarding the impact of the RRH density on the selection of cost-effective communication technologies for C-RANs.

With respect to individual links, the study results showed how sensitive the wireless communication systems addressed herein (MRT and FSO) are to weather conditions. Moreover, the effects of rain and fog were mapped into a cost-effective link solution chart that is dependent on link length and the required bit rate; in this manner, a system designer can be aware beforehand, and without making computations, about which technologies suit better for the intended link and that deserve to be thoroughly surveyed on the market.

Concerning the analysis performed for different RRH densities, it was seen that for few and sparsely distributed RRHs (rural environment), the technology usage distribution to set up a cost-effective fronthaul significantly changed when the weather conditions varied. On the other hand, when considering an area with a high RRH density (urban environment), it was concluded that one technology was extremely dominant when setting up a cost-effective network; additionally, it was concluded that this technology varied between FSO systems, regarding scenarios with good visibility conditions, and FO systems, namely when FSO systems were strongly impaired by the presence of fog.

One of the main findings of this work was that, regardless of the environment type, the performance of FSO systems is a key factor in terms of the total costs of the fronthaul network. More specifically, the possibility of exploiting the FSO technology when there is low fog intensity entails significant savings in the network costs. Accordingly, this shows how it is of utmost importance for a project manager to collect information about the visibility conditions regarding the implementation scenario (e.g., frequency and duration of fog events), in order to get the most out of FSO systems and, consequently, lowering the fronthaul segment costs.

Lastly, it is worth mentioning that any real implementation of an RRH–BBU link requires additional considerations other than the ones addressed herein (such as site availability, rental contract terms, reuse of previously acquired technology, etc.). In spite of that, the provided methodology (namely the supplied software tool) along with the conclusions of this work can serve as important guidelines for fronthaul network designers. For instance, after adjusting the inputs of the aforementioned tool according to the specific situation being analyzed, a project manager will have a better notion of not only which equipment can be disregarded in the first place, thus saving time, but also which situations require further evaluations for a successful network deployment—e.g., if an FSO system is being considered, then it should be investigated whether the Sun or the presence of trees will not impair this system—which also enables to obtain a more productive and time-efficient fronthaul design.

**Author Contributions:** Conceptualization, I.S., N.S., and A.R.; methodology, I.S. and N.S.; software, I.S. and N.S.; validation, I.S. and N.S.; formal analysis, I.S. and N.S.; investigation, I.S. and N.S.; resources, M.P.Q. and A.R.; data curation, I.S. and N.S.; writing, original draft preparation, I.S. and N.S.; writing, review and editing, I.S., M.P.Q., and A.R.; visualization, I.S., M.P.Q., and A.R.; supervision, M.P.Q. and A.R.; project administration, M.P.Q. and A.R.; funding acquisition, M.P.Q. and A.R. All authors read and agreed to the published version of the manuscript.

**Funding:** This work was funded by Instituto de Telecomunicações and by FCT/MCTES through national funds and when applicable co-funded EU funds under the projects UIDB/EEA/50008/2020 and OCTHOPUS.

**Conflicts of Interest:** The authors declare no conflict of interest.

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
