# Peer review of "Fronthaul Design for Wireless Networks"

_applsci, doi:10.3390/app10144754_

Round 1
Reviewer 1 Report
This paper is a well written and structured work on methodology for evaluating and comparing competitor fronthaul technologies for wireless networks. However, due to the fact that there are a lot of approximations for justifying the final results of choosing the most appropriate method, there is a main question arises that the author should reply, whether there is a chance to cut down more approximations and make this methodology more compact without based on so many approximations. Specifically there are approximations in
Lines 217-218
Lines 227-231 and 246-248
Lines 253
Lines 271
Lines 345-348
Lines 351-354
Lines 385-390
Therefore, I think that this paper must be revised again, after the author’s reply to the one major comment and a few minor ones below, as it provides added value on that specific subject and worths for publication afterwards.
Other comments
- Perhaps it would be reasonable for the author at the end of subsection 2.3 to announce its suggested solution, overcoming all aforementioned bottlenecks of those previous related works.
- The author should explain a bit more, lines 230-131, and 247-248 respectively
- Judging form lines 253 and 271, is it always the case for margin value the 3dB?
- Lines 270-273, a reference would be appropriate
- As the author claims that the number of hops are not included in this work but as it seems in line372 there is the option of varying(minimizing) distance, could the latter approach compensate for the lack of considering number of hops?
- Lines 374-376 finally as it can be deduced here, is there an 1-1 relationship between RRH and BBU as claimed previously (in lines 368) or not?
- According to lines 410-411, as the author claims that rain and fog intensities are considered, hence is there a need to include FO option or not?
Reviewer 2 Report
The authors propose a methodology for determining the most cost-effective solution for RRH-BBU links and an algorithm for optimizing the number a position of the BBUs in order to minimize the network costs, in the context of C-RAN architectures. The proposed solutions are tested in various scenarios (different weather conditions, different environments). The article is clearly written and treats a current subject. The authors should consider the following comments and suggestions in order to improve the overall quality of the paper.
Comments regarding the technical contents:
- It is mentioned in the text from rows 206-210 that only single hop-links are considered by the proposed algorithm; it is not clearly mentioned what happens if the required link distance is too long to be accommodated in a single-hop by one of the used technologies; this is a serious limit of the proposed algorithm and it should be commented in more details;
- The comment from rows 249-250 regarding the fact that the same percentage of time for link unavailability regarding the attenuation related to both rain and fog is unclear. Is it normal to have the same rain/fog percentage?
- It is necessary to give and comment the actual expressions that are used for calculating the link budget for the different technologies that are considered and compared for implementing the RRH-BBU links, and the formulas for evaluating the SNR values for different modulation techniques;
- In figure 2 describing the link design algorithm, the question “MRT or FSO technology?” cannot be answered with yes/no, the flowchart should be corrected;
- It is not clear what are the actual values for F.Costs and V.Costs are? Are they inputs or outputs in the .dat files from Table 2?
Comments regarding grammar/typos:
- Consumers instead of costumers (row 22);
- The phrase from rows 132-135 should be split in three shorter sentences;
- First sentence from section 4 is too long and should be split in two;
Comments regarding editing aspects:
- The contributions of the paper, although mentioned in the first section, should be highlighted in a clearer way, by describing the advances over the state-of-the-art brought by the current paper;
- References 7-11 are not commented in the text, some details should be given regarding the communication technologies proposed in that papers;
- Some of the references are missing information regarding page numbers (10,11,32) or the type of paper (6).
Round 2
Reviewer 1 Report
All comments have been replied succesfully, no further comments.